# Interaction between Feet and Gaze in Postural Control

**DOI:** 10.3390/brainsci12111459

**Published:** 2022-10-27

**Authors:** Maria Pia Bucci, Philippe Villeneuve

**Affiliations:** 11MoDyCo, UMR 7114 CNRS Université Paris Nanterre, 92001 Nanterre, France; 2Posture Lab., 7 Porte de Neuilly, 93160 Noisy le Grand, France

**Keywords:** proprioception, vision, foot, eye movements, posture

## Abstract

In the last decade, the number of postural studies in humans, in particular on deficits in postural control in patients, has increased. In the present article, we review studies on postural control with a particular focus on the role of foot and visual inputs in a quiet postural stance. A search and synthesis of articles identified through the electronic databases Scopus, Web of Science, EBSCO, PubMed, and ResearchGate published until 2021 was performed. The aim of this review is to provide reference values for further studies dealing with postural control in both healthy and pathologic subjects and to encourage the development of suitable protocols that take into account the specificity of the different systems responsible for controlling human balance.

## 1. Introduction

Interest in the relationship between the eyes and the feet started in the second half of the 19th century with two German doctors, the neurologist Romberg and the physiologist von Vierordt. Romberg showed that occlusion of the eyes revealed foot sensory disorders [1], while von Vierordt [2] was the first to measure the contribution of sensory information from the visual and foot systems. He showed that postural oscillations of the subject depended on sensations of the muscles, of the sole of the feet, and of visual input. His colleague [3] clarified the role of plantar afferents in postural stability by anesthetizing the plantar sole.

Subsequently, 20th century research focused mainly on the action of gaze and vision on orientation and postural stability. It was only towards the end of the 20th century that the role of the feet in the stability and orientation of the gaze was taken into account.

Baron [4,5] observed a change in the symmetry of the tone of the paravertebral muscles by surgically modifying the tension of the oculomotor muscles in fish. Even blinded, they still presented these postural disorders. Interestingly, he showed that to observe muscle hypertonia, the oculomotor imbalance had to be between 1 and 4 degrees and that beyond 4 degrees the tone of the paravertebral muscles was no longer modified.

In 1963, Held and Hein [6] showed in animals that the development of visual integration in a navigation task was dependent on the movement of the legs. Kittens (between two weeks and ten weeks of age) reared in the dark were subjected to rotatory locomotor activity for three hours. One of the two kittens caused the movement of the other, placed in a basket pivoting around a vertical axis as in a carousel. The first moved actively, unlike the second. Their perception of the visual environment was identical. When the kittens were placed in a real environment, the active kitten behaved normally, while the passive kitten made many errors of visual perception and was unable to avoid obstacles in the environment.

During locomotion, the information from both the legs and the eyes enabled the active kitten to integrate the visual information, while the passive kitten (with immobile legs) was not able to distinguish whether the movement came from the environment or from self-displacement, resulting in perturbed locomotion learning. This is in line with the theory on the memorization and learning mechanism based on visual information suggested by Hebb [7]. Later on, Paillard [8] also reported that the orientation of the legs of kittens depended on the vision experience. In the experiment, a collar that was either transparent, allowing vision of the body and the legs, or opaque, thus eliminating this possibility, was placed around the neck of the kittens. The kittens then had to perform a reaching task with one of their anterior legs. Kittens that did not have the experience of seeing their own legs in motion were unable to visually guide the movement of their legs in extracorporeal space. These two experiments showed that the functional association between legs and vision during development is essential for locomotion and leg orientation. It is well-known that sensory development is initiated in the fetus at the cutaneous level, first of all by the extremities, in particular the mouth, then by the soles of the feet, respectively, at 8 to 11 weeks of post-conceptional age [9].

Lee and Aronson [10] reported that children between 13 and 16 months can fall during a translation of their visual environment. From the age of seven, however, the disturbance of visual information affects postural control less than a foot disturbance. These authors suggested that the most effective mechanosensitivity for standing is probably due to the muscles, the ankle, and the soles of the feet. Interestingly, using a similar experimental paradigm, Bronstein et al. [11] confirmed Lee and Aronson’s results. In addition, they reported that visually evoked postural responses induced by translations from the visual environment can be rapidly suppressed by the cognitive state or simple stimulus repetition. Note, however, that the maturity of vision occurs later in children [12,13], consequently postural control is also under development [14]. The purpose of this review, after briefly reporting the first studies focusing on how the postural system works, is to discuss the role of foot and vision on body stabilities in the quiet stance over the life span of humans; particularly, we consider the interest of studying the interrelationship between vision and feet and their influence on postural sway in a quiet stance.

## 2. Methods

Electronic databases (Scopus, Web of Science, EBSCO, PubMed, and ResearchGate) were searched for manuscripts published up until 2021. We used the following search terms: “postural sway” AND “foot” AND “vision”. We found 98 papers. Some older references were also considered if they were frequently cited and relevant to our aim. Inclusion criteria: (i) language restricted to English, French, Spanish, and German, (ii) publication in peer-reviewed journals. Exclusion criteria: (i) case reports or case series.

## 3. Postural Control and Postural System

Historically, postural control research developed during three phases. Initially, in the 19th century, scientists observed postural and locomotor reactions following the section of a sensory organ in animals or neurological pathologies in humans. Then, at the end of the 20th century, research turned to the weight of sensory organs in closed-loop regulation (feedback), using mechanical devices that disrupted stability. Finally, recent research has also focused on open-loop regulation (feed-forward) linked to an internal model.

First of all, Flourens [15] then de Cyon [16] observed manifest postural modifications after sectioning, respectively, the inner ear, neck muscles, or oculomotor muscles. In humans, the observations related to patients, for example with neurological deficits.

Then, thanks to the input of a biomedical engineer, Nashner [17], closed-loop regulation models dominated studies on postural control. Disturbances in the support (rotation or translation), associated with visual modifications (open/closed eyes, stabilized vision), opened certain loops and thus modified the weight of the main postural sensors (vestibule, eyes, and feet and ankles).

Following this study, Nashner and McCollum [18] described three different equilibration strategies in the sagittal plane. In the first strategy, where travel velocities are low, below semicircular canal reaction thresholds (250 ms) [19], the oscillatory movements are organized around the axis of the ankle joints in which the first muscles to react to the disturbance induced are the extrinsic muscles of the foot, then the thigh muscles and then the trunk muscles. The motor sequence is disto-proximal and has been referred to as the “ankle” strategy, found in young, healthy humans with minimal disturbances to the base of support.

Another strategy is organized around the axis of the hips. This strategy is found in some patients and some seniors, or during significant exogenous postural disturbances. Woollacott et al. [20] showed that the equilibration reactions of the elderly are different from those of young subjects. They use more of the “hip” strategy. It is likely that its use is the result of a failure to use the “ankle” strategy [21]. The third strategy is organized in relation to the vertical axis in which we find not only ankle and hip flexion, but also knee flexion.

From a physical point of view, postural stabilization can come from two endogenous mechanisms: the displacement of the center of plantar pressure (CPP) with respect to the vertical projection of the center of mass (CoM), which corresponds to the “ankle” strategy, or opposite segmental movements around the CoM, which corresponds to the “hip” strategy [22].

To determine the respective weight of each sensory input, Fitzpatrick and McCloskey [23] designed an ingenious device to analyze, respectively, the visual, vestibular, and foot (proprioceptive) systems and to measure the perception thresholds of each of these inputs reported by healthy subjects during very small movements. Standing with eyes open, the subjects were able to perceive movement variations from 0.17° when the speed was slow (0.06°/s). At higher speeds (0.11°/s), all subjects perceived movements from 0.11°, sometimes smaller (0.06°). On the other hand, the vestibular system played no role in the perception of the usual oscillations of the body during the standing position. This research showed that sensory inputs do not all have the same minimum threshold of perception. For low speeds, close to those of natural oscillations, only the foot system works whereas for slightly higher speeds, vision contributes to postural control [23]. The threshold of the vestibular sensors is greater than the variations linked to postural oscillations in the easy stance [19,24]. The vestibular system controls trunk orientation [25] and is related to the hip strategy [26].

Interestingly, Honegger et al. [27] explored the interaction between the head, the trunk, the pelvis, and the ankles in healthy subjects and vestibular patients, in the quiet stance, and confirmed that, depending on the oscillation frequencies, the postural behavior could be modified. At low frequencies (<0.7 Hz), the head was locked to the trunk, following the inverted pendulum model. At medium frequencies (about 3 Hz), the head moved slightly in antiphase to the stabilized trunk, stabilizing the gaze. At high frequencies (>3 Hz), the movement between the head and the trunk was in phase, but not between the head and the pelvis (anti-phase), evoking a “hip” strategy. In addition, they specified that healthy subjects and vestibular patients showed more movement in the ankle and leg than in the pelvis; given that their head moved more than their trunk, they were in the presence of a partially damped pendulum [28]. In summary, in postural conditions where the base of support undergoes no, or only minimal (low frequencies, <0.1 Hz), disturbance, subjects oscillate similar to a damped inverted pendulum, which stabilizes their head and their gaze. In contrast, during faster disturbances (>0.5 Hz), vision intervenes to stabilize the head. At even higher speeds (1.0 and 1.25 Hz) the vestibular system intervenes and stabilizes the head in space, and the inverted pendulum becomes multisegmental (“hip” strategy). Visual information is important to maintain a fixed position of the head and trunk in space, while proprioceptive information is sufficient to produce stable coordination patterns between the support surface and the legs (see the following sections).

## 4. Role of the Foot System in Posture

The foot system is the interface between the body and the ground. Consequently, it plays an important role in postural control, orientation, and locomotion.

The different muscles of the foot actively participate in mobilizing the feet and keeping the body in the axis erect, by stabilizing the leg, the base of the human inverted pendulum. Initially, only the extrinsic muscles of the foot were investigated electromyographically; for instance, Okada [29] showed that, when standing in a quiet stance, the maximum tonic muscle activity was due to the soleus, while the rectus femoris and vastus lateralis exhibited less activity. Then, Schieppati et al. [30] highlighted the role of the intrinsic muscles of the foot, especially the flexor digitorum brevis. Voluntary tilting of subjects forward increased the soleus and the flexor digitorum brevis activity, suggesting a role of these muscles in fine-tuning postural adjustment during forward tilt [31]. When moving at moderate speed from the base of support, the muscles of the feet stabilize the head [32]. Intrinsic feet muscle activity increases with postural activity and contributes significantly to the maintenance of postural balance [33].

When moving the base of support backwards, for accelerations below the threshold of the vestibular system, the majority of subjects show an initial reaction of the extrinsic muscles of the foot (90 to 100 ms), then 10 to 20 ms later by the thigh muscles and another 10 to 20 ms later by the trunk muscles. This muscle reaction sequence is disto-proximal [18], but the motor sequence can be reversed and becomes proximo-distal in the elderly [34], neurological patients, and healthy young subjects with a disturbance of their base of support or anesthesia of their feet [35,36]. In this case, the head is no longer stabilized, the trunk tilts forward, the vestibular system intervenes in postural control [26], and the muscles of the foot system participate much less in postural regulation [36]. When foot sensory information is disturbed, the weight of the vestibular system increases [25], but patients with profound bilateral vestibular loss showed similar patterns and latencies of leg and trunk muscle responses to body movements as healthy subjects [32].

Babinski observed as early as 1899 [37] that a healthy man, when asked to tilt his head and trunk backwards, also performs a bending movement of the knees. These coordinated unconscious muscle activities that precede voluntary movements and counter associated postural disturbances are called anticipatory postural adjustments (APA) [38,39,40].

Belen’kii et al. [41] found that when a standing subject is asked to move his arm in the sagittal plane, he first executes a muscular activation of his legs, then of his trunk, and then mobilizes his arm. This observation has since been confirmed and clarified by many authors [42,43,44,45]. When performing a voluntary movement such as pulling with the hand on a bar attached to a wall, subjects activate the ankle extensor muscles (gastrocnemius) 100 ms before the arm flexors [43]. Other anticipatory neuromuscular activities have been studied, in particular during leg elevation movements, head rotation [46]), rotation of the eyes [47], and even in relation with reflex visceral movements, such as coughing [48].

These posture adjustments must be planned in advance by the CNS and a predictive neural control mode sends commands to the muscles to initiate and stabilize the posture. They are considered to be predictive in nature because they are produced before feedback from the current movement can influence them [39,49]. The central commands of APAs appear in different regions of the cortex [50], but the brainstem and the reticulospinal system play an important role in postural adjustments [51]. Neurons in the pontomedullary reticular formation (PMRF) fire strongly in response to unexpected disturbances and in a manner consistent with contributing to compensatory responses that restore balance [52]. Damage to the reticulospinal system impairs balance [53]. Signals from the PMRF contribute to predictive postural adjustments [54,55]. The architecture and physiology of the foot seem to contribute to the task of postural control with great sensitivity [56]. Previously, Roll [57] had highlighted a proprioceptive chain linking the feet and the eyes. Tendon vibratory stimulation of the extrinsic muscles of the foot generates an illusion of movement of a visual target, while the stimulation of the oculomotor muscles modifies the orientation of the center of pressure [58].

Hollands et al. [59] were the first to focus on the movements of the whole body, related to the orientation of the gaze in the horizontal plane. They studied the coordination of eye, head, trunk, and foot movements in a rotation task of up to 135°. They objectified that the most important correlation was established between the gaze and the feet. The multi-segmental coordination that unites the eyes to the feet depends on the CNS which allows stabilization of the body.

In addition to these biomechanical aspects of the foot, the weight of the foot sensors, particularly the cutaneous ones, are also implicated in postural control. It has been shown that the absence of vestibular and visual cues has little effect on postural stabilization [60], confirming in a calm position the predominance of signals from the foot system in postural control. To advance in understanding the role of the foot system, it is necessary to dissociate exteroceptive information, coming from plantar skin receptors, from those related to proprioception, coming from muscles and joints.

Over the last twenty years, publications on the skin have enabled an understanding of its importance in postural, non-nociceptive control (for a review see [61]). Many types of non-nociceptive plantar skin stimulation alter postural regulation. For example, Okubo et al. [62] showed that small pellets emerging from a 1 mm board, distributed under the entire plantar surface, reduced the oscillation surface, but only with open eyes. At the same time, clinicians [63,64] proposed that thin reliefs (1–3 mm) placed on soles could help relieve patients with chronic non-specific back pain. Given their objective, Villeneuve [65] called them posture insoles. More recently, researchers have confirmed the role of low-intensity plantar biomechanical stimulation. Kavounoudias et al. [66] showed that vibrational stimulation of less than 1 mm under the soles of the feet led to whole-body inclination in all subjects and that the direction depended on the areas of the foot stimulated and was always opposite to the pressure increase simulated by the vibrations. Plantar input contributes to the coding and spatial representation of body posture [67]. It has a role of a “dynamometric map” [66]. Recently Kenny et al. [68] showed that textured insoles with 3 mm thick reliefs decreased anteroposterior oscillations. Low-thickness reliefs used in the manufacture of posture insoles have shown their effectiveness on postural control [61,69,70,71] and oculomotricity [69,72]. Viseux et al. [61] showed that 0.8 mm bumps had a significant effect on control, but that 6 mm bumps did not. Reliefs that are too thick beyond 3 mm could saturate the skin sensors [73]. The reliefs used in posture insoles are likely to modify postural control, eye movement, and cortical activity [74].

## 5. Role of Vision and Eye Movements in Posture

Lee and Lishmaan [75] first described the importance of vision for controlling body sway in the natural environment and its dependence on the task, the age of the subject, and any pre-existing pathology [76,77,78,79,80]. Under normal conditions, it is well known also that subjects with their eyes closed are more unstable: the surface of the center of pressure (CoP) is about 2 or 3 times larger with eyes closed with respect to eyes open (the so called Romberg Quotient, that is the ratio between the postural parameter obtained with eyes closed/the values obtained with the eyes open). It is also well known that an important effect of the distance of the object from the subject is fixation. When the target is distant, the amplitude of the surface of the CoP is larger; consequently, the subject is more unstable than when he/she fixates a target at near distance. This has been observed in adult subjects [81,82,83,84] as well as in children [85]. All these authors agree with the hypothesis of a decrease in visual motion signal due to a geometric effect; that is the angular size of the retinal slip is greater in near vision than in distant vision and its detection is more difficult in far vision, leading to a less efficient correction of body oscillations. Additionally, in addition to the angular size of visual inputs, the signals from extraocular muscles related to vergence (convergence or divergence eye movements) could have an influence on body sway [86].

Given that vision is one of the most important inputs to control posture, the question arises as to the quality of body sway in subjects with visual impairment. Friedrich et al. [87] examined postural control in normal adults and in adults with visual disabilities and reported that the latter were able to compensate their visual deficits by activating other processes (via the cerebellar network) to ensure good postural control. This finding is consistent with the study by Assländer and Peterka [88], which showed that healthy adult subjects were able to reweight different types of sensory information in order to maintain their stability.

Note, however, that these adaptive mechanisms do not always work correctly. Harwood [89] reported that visual impairment decreased postural control, especially in older adults, for whom poor postural stability is the main risk factor for falling [90,91]. Lamoureux et al. [92] studied the impact of visual impairment on the risk of falling in adults aged between 40 and 80 years with moderate or severe visual impairment in one or both eyes and compared the results with those of age-matched controls. They demonstrated that severe visual impairment in the worse eye increased the risk of falling and that severe visual impairment in one eye and moderate visual impairment in the other eye doubled the risk of falling. In line with these findings in a study of our group [93], we reported that subjects with Age-related Macular Degeneration (AMD) were more unstable than healthy age-matched control subjects (with increased anterior–posterior displacement of the center of pressure), most likely because of their poor visual capabilities and poor adaptive mechanisms that were unable to compensate their visual deficits.

Strabismus also affects postural stability negatively. Matsuo et al. [94] were the first to focus on the important role of binocular vision in postural control. They measured the displacement of the CoP in 17 adults (mean age: 39 years old) with and without stereoscopic capability and reported that subjects with stereoscopic vision were more stable than subjects without it. The first studies on postural control in strabismic children appeared in Sweden in the mid-1980s. Initially, it was shown that children with convergent strabismus were more unstable during walking than children without strabismus [95]. The following year, Sandstedt et al. [96] showed that children with divergent strabismus were less stable than children with convergent strabismus. However, these studies did not examine objectively the displacement of the CoP. It was not until 2006 and the work of Matsuo and collaborators [97] that the displacement of the CoP in strabismic children was measured. The authors recorded postural sway in strabismic children with and without stereoscopic vision and found that children without stereoscopic acuity were more unstable than children with stereoscopic vision, suggesting the importance of visual input in the postural regulation process. Studies by our group [98,99] also reported that strabismic children were more stable when the non-squint eye was viewing, highlighting the important role of binocular vision for postural control in these children. We also reported that eye surgery improved postural stability in strabismic children [100]. All these studies are in line with the hypothesis that the compensatory mechanisms mentioned above could be still immature in children, leading to difficulties in correctly using the other sensorial inputs to control posture.

Several researchers have investigated the impact of eye movements on postural control by recording CoP excursions on a platform. Interestingly, different effects have been reported, i.e., degrading, having no effect, or even reinforcing postural stability, most likely due to the different experimental conditions tested (different types of eye movements elicited, different types of postural condition tested, bipodal or unipodal position, etc.). For instance, Brandt et al. [82] and Brandt [101] found a decrease in postural stability when performing saccades greater than 40° compared to a fixation task. The authors suggested that beyond 40° this effect was due to the high retinal speed produced by larger saccades. In contrast, White et al. [102] failed to show any effect of saccades of smaller amplitude (4 deg) on postural performance (standing on one foot) and even an improvement in body stability while performing horizontal as well as vertical saccades [103,104,105]. Stoffregen et al. [106] reported a reduced sway while subjects were performing saccades with both eyes open as well as eyes closed in line with their hypothesis of a functional integration of postural control with visual performance. In other words, to execute saccades correctly, postural control could be modulated. In line with this hypothesis, Rougier and Garrin [107] compared the effect of blinks and saccadic eye movements on postural stability in adults and they found that blinks did not change postural sway but that horizontal and vertical saccades reduced it, reinforcing the supra-postural concept of Stoffregen et al. [106], who argued that postural control could change in order to facilitate the performance of the oculomotor task. Few studies have explored the effect of saccades on postural stability in children. Our group measured postural control in 10-year-old children while performing saccades or reading a text [108]. We found that saccades improved postural stability in comparison with reading. This could be due to the fact that during reading saccades are done together with other cognitive activities (namely word comprehension), demanding a focus of attention on the reading task and leading to poor postural control in children.

The relationship between pursuit eye movement and body sway has been also explored in several studies. For instance, Baron et al. [109] reported that the pursuit of a target oscillating at 0.3 Hz increased the CoP displacement. Brandt’s group also investigated the effect of pursuit eye movements on postural control by comparing it to the effect of a simple fixation task [82,110,111]. The results showed that in a stable or mobile visual environment, the eye movements caused by pursuit eye movements induced an amplification of postural oscillations. They explained this finding using the extra retinal signals resulting from the pursuit movement, used by the central nervous system during the motor control of the body oscillations, which affected postural stability.

It should be pointed out that, in all these studies exploring eye movement performance and postural control, researchers measured body sways only, without recording eye movements. The first study, to our knowledge, that recorded both eye movement and postural sway was the study by our group [112] in which both fixation and saccades and posture were recorded with an eye tracker and a platform in a large group of healthy children (from 6 to 17 years old). Postural sway decreased significantly while children were performing saccades in comparison with a simple task of fixation and there was a general improvement in both postural and oculomotor performance with the age of the children. The interaction observed between oculomotor and postural abilities could be due to the fact that the same structures of the central nervous system play an important role in postural control as well as in programming and executing eye movements [113]. Additionally, the postural improvement observed by age of children could be due to perceptual, motor, and socio-emotional capabilities arising from experiences that naturally occur during the developmental process. Other studies conducted by our group [114,115,116] recording at the same time both eye movements and postural sway reported that while performing pro- and anti-saccades (respectively, saccades toward and in the opposite direction to a target), the children significantly reduced their body displacement, while during pursuit eye movements the children became more unstable.

Taken together, all these studies suggest that the influence of oculomotor tasks on postural control differs depending on the attentional demand of the task. In order to explain the effect of a secondary task on body control, several models have been proposed to explain how attention can be focused on the postural task or on the other secondary task. Attentional capacity has to be distributed between the two simultaneous tasks in various ways, according to the strategy chosen by the subject. The more demanding the main task is, the more attention must be allocated to that task to maintain an acceptable level of performance. Attention has to be divided between postural control and the secondary task; consequently, postural performance is diminished in a dual-task situation compared to a single-task situation.

Lacour et al. [117] summarized the three main relevant models capable of describing the inter-relation between attention resource and postural control. The cross-domain competition model postulated that posture control and the secondary task compete for attentional resources so that postural performance in dual-task conditions is always altered compared to the single postural-task performance. Because of attentional resource sharing, balance performance will be less efficient in dual-task conditions. In contrast, the task prioritization model is mainly found in the elderly to avoid a fall. When the situation for maintaining posture is difficult (dynamic platform, reduction in support), the subject is able to change postural strategy to obtain a good balance. The modes of allocation of attentional resources can also differ according to age. The “posture-first” hypothesis predicts that older people focus more on their postural control than on their secondary performance to avoid falls.

Finally, the U-shaped non-linear interaction model described by Huxhold, Li, Schmiedek, and Lindenberger [118], suggested that the secondary task could either increase or decrease postural stability depending on the type of task, and on the attentional cost of the task. For instance, fixation and pursuit eye movements are quite difficult attention-demanding tasks, leading to degradation of the postural sway. In contrast, an easy task, such as performing saccades to a target, shifting the attentional focus away from postural control, leads to a better automatic postural performance.

## 6. Discussion

In this article we have reviewed current research on postural control in humans, focusing particularly on the relationship between the different sources of sensory information that are important in controlling body sway in natural environments. We observed that the role of each of these sensory inputs (visual, vestibular, and foot) in controlling posture is different in the human population; in children, vision is the most important input to maintain a good level of body balance, whereas with age proprioception influences postural sway more given that the other sensory inputs are degraded in the elderly.

Our goal was to underline the need to develop systems in order to measure simultaneously proprioception, foot, vestibular, and vision inputs in the control of posture for healthy subjects as well as for patients. As described above, we reported that an eye tracker is essential to verify what the eyes of the subjects are really doing during the experiment in which subjects are asked to move their eyes by performing saccades, pursuits, fixations, and so on. On the other hand, eye tracker systems need to compensate (and to measure) head movements given that, as we reported, the head position affects postural control. The eye trackers used in the above-reported studies were not able to record head movements; consequently, the analyses of eye movement parameters were incomplete since measurements of the amplitude, precision, and velocity of eye movements were lacking. This review highlighted the interaction between the two extremities of the postural system, the feet, and the eyes which code the information coming from the environment: the base of support and the extrapersonal space. Note also the foot system contributes to the biomechanical and sensory aspects of postural regulation; indeed, in a calm position, the muscles of the foot system are the most active and, when external postural disturbances are absorbed, they are the first to react. This allows them to stabilize the head. They also initiate APAs during voluntary controlled lower limb arm movement and even the gaze. Moreover, during the gaze orientation task, requiring coordination of the whole body, the foot is the segment best correlated with the gaze. From a sensory point of view, subtle plantar cutaneous stimulations modify postural control and oculomotor performance. These interactions between the foot and the gaze demonstrate the coupled nature of visuomotor and postural control.

Furthermore, we pointed out the role of vision and of eye movements on postural sway taking into account attention focused on performing a double task. Based on all the studies described above, we can assume that under a simple postural task, adaptive mechanisms exist to obtain good postural control, but under a more complex postural task or in a dual-task situation, compensation is more difficult to achieve, particularly for children, leading to body instability. The difficulty of the type of eye movements executed during postural control and their effect on attentional demand also affect body sway. The role of attention in postural sway is particularly important in subjects who are still developing and in the elderly, given that at these ages the central nervous system is not completely developed (for young children) or is starting to decline (in the elderly).

Of particular interest are the findings on the interference between vision, particularly eye movements, and postural control given that in natural conditions, humans naturally move their eyes during walking, sitting, standing—in other words, a human being is always in a dual-task condition. In these conditions, the attentional cost required for performing the two tasks modulates such an interaction, leading to cognitive processes and sensory-motor adaptations.

In the future, researchers will develop protocols in which each sensory system can be controlled and changed in order to distinguish their role in age-dependent and/or pathology-dependent disabilities of postural control. The literature reviewed in the present article will help the reader to have an overview of how balance is achieved in healthy humans and how adaptive mechanisms are able to avoid imbalance or falls.

Finally, this knowledge will be useful for developing the therapeutic rehabilitation of patients in order to improve their postural control system and to prevent an increase in balance decline with age.

The relationship between the different multisensory inputs in postural control will also need to be taken into account in exploring sensory reweighting in response to variations observed in subjects with neurodevelopmental disorders such as autism, cerebral palsy, attention deficit/hyperactivity disorder (ADHD), and so on, as all these disorders are characterized by motor deficits, and in particular postural instability.

Further research needs to be done to improve knowledge on the interaction of the different sensorial inputs on body balance control; for these kind of studies, specific systems need to be developed that allow for the easy recording and measuring of all these sensorial inputs simultaneously in order to observe the eventual specific deficits and eventual postural adaptations. This will allow researchers and scientists to expand the fundamental knowledge on postural control and clinicians to take advantage of these technologies for guidance in further studies and to design specific interventions for improving balance in patients. Furthermore, the results of this review point toward a systemic understanding of the eye-foot relationship with postural control and should be further taken into account to optimize therapeutic results in the context of chronic pain and cognitive disorders. For instance, clinicians have previously shown relationships between hypertonia, pain, and eye movement.

Finally, we have to point out that in the present work we reviewed articles on quiet standing postures only. Posture quality during dynamic conditions needs to be deeper discussed in further work.

## Data Availability

Not applicable.

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
