# Peer review of "Interaction between Feet and Gaze in Postural Control"

_brainsci, 2022, doi:10.3390/brainsci12111459_

Round 1
Reviewer 1 Report
Authors provide an interesting literature review on postural control, focusing on the relationship between foot and gaze. Even if the topic is interesting and the provided overview is clear, I strongly believe that the structure of the paper does not meet with the requirements of a literature review. In fact, paper completely lacks of information related to search strategy, paper quality evaluation and paper selection. These are mandatory aspects to consider when writing a systematic review.
Please add a materials and methods section and check other systematic reviews to understand the missing information.
Author Response
Reviewer 1: Authors provide an interesting literature review on postural control, focusing on the relation-ship between foot and gaze. Even if the topic is interesting and the provided overview is clear, I strongly believe that the structure of the paper does not meet with the requirements of a literature review. In fact, paper completely lacks of information related to search strategy, paper quality evaluation and paper selection. These are mandatory aspects to consider when writing a systematic review. Please add a materials and methods section and check other systematic reviews to understand the missing information. We added some information in a short Method section. Note, however, that other ‘re-view’ articles public in Brain Sciences or in other journals did not detailed methods used to selection of papers revised Quintero, et al. Brain Sci. 2022; Premeti et al. Brain Sci. 2022; Wang et al. Brain Sci. 2022; Vieux et al. Brain Sci. 2022; Wittling et al. Front Immunol, 2021; Reviewer 2: Well done review about the most important studies concerning interactions between visual system, proprioceptive system, vestibular system and CNS. Prospects for further studies by the authors seem to be promising. Thank you very much for your commentsReviewer 2 Report
Well done review about the most important studies concerning interactions between visual system, proprioceptive system, vestibular system and CNS. Prospects for further studies by the authors seem to be promising.
Author Response
Reviewer 2: Well done review about the most important studies concerning interactions between visual system, proprioceptive system, vestibular system and CNS. Prospects for further studies by the authors seem to be promising. Thank you very much for your commentsReviewer 3 Report
ABSTRACT
Ampliated the abstract in order to understand the complete research conducted
INTRODUCTION
The introduction must provide sufficient background information for readers to understand the research aim. the authors should clarify the importance of this topic and the actual knowledge in this area. It seems like some important clarification is missing.
Include a concise objective of the study and describe methodology used to reach the aim.
DISCUSSION
Must be ampliated to give to the reader a deeper point of view of the thematic
Conclusion should respond the research aim
Explain limitation of the study and future research line according to the study conclusion
Include a practical application section
Author Response
Reviewer 3: ABSTRACT Ampliated the abstract in order to understand the complete research conducted Complied INTRODUCTION The introduction must provide sufficient background information for readers to understand the research aim. The authors should clarify the importance of this topic and the actual knowledge in this area. It seems like some important clarification is missing. Include a concise objective of the study and describe methodology used to reach the aim. According to your suggestion we improved the Introduction and a short Method section was also added DISCUSSION Must be ampliated to give to the reader a deeper point of view of the thematic Conclusion should respond the research aim Explain limitation of the study and future research line according to the study conclusion Include a practical application section Discussion was improved following your suggestion and limit of the study and further research projects were addedRound 2
Reviewer 1 Report
I have no further comments.